# Physical Exercise and Older People: Always a Happy Relationship? Four Qualitative Reflections to Deepen Understanding

Alexis Sossa Rojas

Institute of Sociology (ISUC), Pontifical Catholic University of Chile, Santiago 8331150, Chile; apsossa@uc.cl

**Abstract:** In this paper, I recall reflections from and discussions with both older people who exercise actively and with personal trainers who specialise in working with older people to address two essential elements that should be clarified: First, what are we talking about when we discuss sport, physical exercise and physical activity, especially when we relate them to older people? Second, the benefits of exercise are known, but what are the margins and precautions that this group of people should consider, and even the damage that physical exercise can cause to them? Based on qualitative data that are taken from different ethnographic works, four areas are considered: What does it mean to train as a senior?; are injuries inevitable?; the dangers of having an athlete's identity; and the hazards of body-image ideals. This work gives voice to older athletes and their coaches, and contributes to studies on physical activity, older people and wellbeing.

**Keywords:** injuries; qualitative methodology; training; older adults; sport





## 1. Introduction

In this work, I recall reflections, stories and discussions I have had with older people who actively exercise and trainers who are specialised in training older people. It is indisputable that physical exercise brings health benefits. However, the literature does not clarify two critical elements as it relates to older populations. Firstly, what do we mean by sport, physical exercise and physical activity, especially when we discuss them in relation to seniors? Second, what margins and precautions should older people take, and what damage can physical exercise cause to this population?

Older adult participation in sports has traditionally been neither expected nor encouraged (Gard et al. 2017); it was even considered at one time that physically strenuous sports could be harmful and life-threatening for the ageing body (Coakley 2017). Older people were cautioned against undertaking sports. After all, rest was once considered a privilege of old age (Grant 2001; O'Brien Cousins 2000).

Nevertheless, some decades ago, exercising and sport began to be heavily promoted for older people. The World Health Organisation (WHO) has launched a global action plan to reduce physical inactivity by 10% by 2025 and 15% by 2030 (Habib et al. 2022). Exercising for older adults is beneficial for many reasons: socially (Jenkin et al. 2018; Lechner 2008); for physical and mental health (Lewis et al. 2017); to improve cognitive function (Boutcher 2000; Daskalopoulou et al. 2017; Langhammer et al. 2018); to boost satisfaction with life and self-esteem during ageing (Oliveira et al. 2018); to prevent and treat chronic disease (World Health Organization 2018); to enhance the immune system (Martin et al. 2009); to improve postural performance (Perrin et al. 1999); to prevent various degenerative diseases, such as sarcopenia and dementia (Yamamoto et al. 2022); to build a sense of self-competence (Reed and Cox 2007); to improve sleep (King et al. 1997); to reduce mortality risk by about 35% (Bull et al. 2020); to postpone disability by 7.75 years (Caprara et al. 2013); to boost older people's sense of wellbeing (Grant 2001); to decrease the risk of development

of various types of cancer (Milton et al. 2014); and to maintain social connections and networks (Active Ageing 2018; Stenner et al. 2020).

Sports involvement in older adulthood has been linked to several psychosocial outcomes. These include assistance with the management of an ageing identity; the challenge of traditional stereotypes of older people; the enhancement of motivation related to continued participation in physical activity; the provision of opportunities to perceive competence or self-confidence, break records and test personal abilities; the gain of opportunities to travel; and the encouragement to develop or strengthen social relationships (Baker et al. 2010; Gayman et al. 2017b). Participation in sports enhances older adults' life satisfaction, social life (e.g., camaraderie, social networking, social belonging, a sense of community) and personal psychological status (personal empowerment, self-worth, self-esteem, self-efficacy, pride) (Eime et al. 2013).

Whether within a team or individually, sports involvement can provide a social network of like-minded people through affiliation with a sports club. This is particularly useful for ageing adults with fewer avenues to seek out social interaction (Nicholson 2012). Finally, sport represents a holistic health tool that has been shown to build at least three pillars of lifestyle medicine: physical activity, management of stress and the promotion of positive social connections (Walsh et al. 2022). It may also help with the other three (healthy eating, restorative sleep and avoidance of risky substances).

Physical inactivity is considered one of modern society's most important public health problems (American College of Sports Medicine 2009; Pedišić et al. 2011). However, despite current recommendations and the known benefits of exercise, only 26% of individuals aged 65–74 years and 10% of those aged 85 and older meet public health recommendations (Lübcke et al. 2012). Recent statistics indicate that older adults' inactivity rates are 19–25% of adults aged 60–69 years and 42–59% of adults aged 80 years and older (Pinheiro et al. 2022). Physical activity is a crucial driver of healthy ageing, and sport provides physical activity in a playful, enjoyable setting.

It is essential to promote the value of sports, while ensuring that partakers understand the limits and precautions to follow so that they avoid the damage an older person can do during sports or physical exercise. It is essential to give a voice to older adults themselves and to their specialist trainers to understand what they regard as sports, physical activity and physical exercise and whether they are the same, especially considering that quantitative research sometimes measures these without a clear distinction between them.

## 2. Some Definitions Problematised

For coaches, older adults and researchers, discussing and problematising some ambiguous definitions is important. The terms "physical activity", "sport", "exercise" and "serious leisure" are sometimes used interchangeably.

Sport is defined as follows:

- All forms of physical activity that, through casual or organised participation, are aimed to express or improve physical fitness and mental wellbeing, form social relationships or obtain results in competition at all levels (Council of Europe 2001);
- A human activity that involves physical exertion and skill as the primary focus of the activity, with elements of competition in which rules and patterns of behaviour governing the activity exist formally through organisation (Australian Sports Commission 2009);
- Sport involves physical exertion and is typically carried out during leisure time, but it is also characterised by elements of competition and skill (Australian Sports Commission 2009).

Guttmann (1978) asserted that play forms the foundation of all sports. This includes voluntary, intrinsically motivated activities that are performed for fun or enjoyment. Suits (2007) believed that a characteristic that distinguished sports from games was that, while sports were games of skill rather than chance, the skill had to be physical, and sports had to consist of physical contests (Tamburrini 2000). Sports were stated to be all goal-directed activities that adhered to rules, and they had to include competition that

resulted in winner(s) and loser(s) (Drewe 2003; Guttmann 1978), in which chance or luck was not the sole reason for winning (Suits 2007).

For those who do not take up sports professionally, sport is often seen as a trivial pursuit, categorised under "hobbies" or things to do once the serious business is over (Peña et al. 2021). Sport-based activities involve movements with defined goals based on explicit formal rules and structured relationships between participants and athletes (Contreras-Osorio et al. 2022). Nevertheless, the reasons why some older adults continue to play sports are not well understood (Stenner et al. 2020).

These definitions contain complexities. What do we mean by skills? What is not a physical activity? How is sport different from a sport-based activity? Questions like these have led to various reviews that have concluded that definitions and methods used to measure sports lack consistency (Gayman et al. 2017b).

Physical activity is defined as any bodily movement produced by the skeletal muscles that substantially increases energy expenditure over resting conditions (Caspersen et al. 1985). In contrast, exercise is any form of physical activity undertaken by an individual during leisure time and performed repeatedly over an extended period to achieve a desired aim in line with improvement in fitness and health-related outcomes (Caspersen et al. 1985).

As with the definition of sport, problems arise here too, because physical activity encompasses complex behaviours that may occur in different settings and modes, and at varying frequencies and intensities. This complexity makes it difficult to measure in a general-purpose social survey (Berger et al. 2005). Likewise, inconsistent definitions of exercise and competitive sport make it impossible to group these differences according to specific physical activity levels (Winterbotham and du Preez 2016). Sometimes, in studies of physical lifestyle in the older adult population, the activities of different people are compared. These activities include walking, gym-going and gardening. There are undoubtedly differences among these, and walking may be a physical activity or physical exercise depending on the condition of the person who does it; gym-going is more related to physical exercise but whether or not it is exercise depends on what the person does at the gym, and gardening is more related to physical activity.

Sometimes, people refer to a particular activity in leisure time as physical activity or exercise. The framework then relates more to serious leisure, which is distinguished from casual leisure based on six characteristics: (a) the need to persevere in the activity, (b) the development of a leisure career, (c) the need to put in effort to gain skill and knowledge, (d) the gain of social and personal benefits, (e) the entry into a unique ethos and social world, and (f) the development of an attractive personal and social identity (Kim et al. 2020).

The problem, then, is not only that the definitions are unclear but that in the particular case of working with older adults, these concepts, if measured, prescribed and used interchangeably, can generate problems in the study of this population and the understanding of the relationship among physical activity, physical exercise and older people.

## 3. Some Adverse Outcomes of Sport and Exercise

The benefits of sports participation later in life are not absolute. The performance of research has been advocated to improve understanding of the potentially adverse individual outcomes of older adults' involvement in sports. These outcomes include sport-related injuries, immune dysfunction, exercise dependency, negative effect and poor ability to cope with an ageing identity or career termination (Baker et al. 2010). Some research has acknowledged that sports participation is associated with costs and benefits in older adulthood (Gayman et al. 2017a). In this article, I discuss four critical topics related to the risks of exercising frequently, which were unveiled during my conversations with my informants and observations during my various fieldwork.

## 4. Methodology

Attending gyms has been part of my academic curiosity and one of my favourite recreational activities. I have been training in different sports since I was 12 years old,

but mainly in gyms, and topics of training, nutrition, injuries and sports achievements, to name a few, have been a constant presence in my readings and my daily life.

This background is important because, with this, I seek to point out that in all these years of training, I have been able to experience the achievements, frustrations, harms and accomplishments that gym training entails first-hand. This discussion is meaningful because, when older people make mistakes in sports or do not know certain things about physical activity, they may cause themselves more damage than younger people in the same position due to the fragility that their bodies and minds tend to show. Furthermore, the health benefits for older people who engage in physical activity have been evidenced predominantly through quantitative studies and biomedical models (Colcombe and Kramer 2003; Hollmann et al. 2007; Muscari et al. 2010). It is indispensable that the qualitative aspects of this relationship be shown and that its protagonists be given their voice.

During my years of training and then as a researcher, I have seen negative situations concerning sports, such as discrimination, substance abuse, injuries, accidents and self-image distortions. After working with older adults who exercise, I felt the need to write about the complexities and negative aspects that training can bring.

Consequently, the data presented here are from different studies and are all from ethnographic work with older adults who train frequently, and from conversations and interviews with personal trainers who specialise in working with older adults. Specific characteristics of the informants in the different fieldwork can be seen in different articles and a thesis (Sossa 2013, 2017, 2020, 2022). I have kept the same pseudonyms for follow-up purposes to quickly identify them.

In each study, the data were anonymised; the informants knew that they were taking part in the studies, and they all gave me their signed consent. In addition, all the studies were carried out according to the ethical principles established by the ethics committees of the various universities with which I was affiliated, and each was approved by the relevant university's research ethics committee.

It is important to note that my research dealt with the benefits of physical activity. However, the negative aspects of sport and training emerged during the fieldwork and conversations conducted for each study. I collected the data that I have now systematised and analysed in this article.

I have worked with many older adults, but in this article, I focus on those with whom I have spoken the most about the negative aspects of physical exercise.

- Francisco was 67 years old when he participated in my study. He was a Chilean man who had competed in seven marathons.
- Claudia was 63 years old when she participated in my study. She was a Chilean woman who was working as a chi kung and Pilates instructor and who had become a kinesiologist and ballet dancer.
- Sandra was 63 years old when she participated in my study. She was a Chilean woman who was working as a physical training instructor and who had also become a physical education teacher.
- Gilda was 68 years old when she participated in my study. She was a Chilean woman who cycled and frequently went to the gym.
- Alex was 72 years old when he participated in my study. He was a Dutch man who frequently went to the gym.
- Todd was 66 years old when he participated in my study. He was a Dutch man who frequently went to the gym.
- Mara was 67 years old when she participated in my study. She was a Surinamese woman who frequently went to the gym.

## 5. Results and Discussion

### 5.1. What Does It Mean to Train as a Senior?

In the literature, different positions are taken up from which to talk about training as a senior, particularly in relation to competitive activity. For instance, a "master

athlete" is defined as an individual over the age of 35 years who competes regularly in state, regional, national or international age-group competition (Appleby and Dieffenbach 2016; Huebner et al. 2022), while "senior athletes" are aged 50 years and older (Cannella et al. 2022). However, an older adult is defined by the United Nations as someone over 60 years of age, yet traditionally, an older person has been defined as older than 65 years. These definitions have led to the definitions of different age ranges in studies (or to the grouping of people of different ages). Those aged between 65 and 74 years are considered early elderly, while those over 75 years are referred to as late elderly. Similarly, definitions such as young-old (60 to 69 years), middle-old (70 to 79 years), very old (80 years and older), or young-old and old-old, are used.

In my research, I have found these definitions to be a challenge. For instance, one set of criteria for classification is that the person should be over 65 years of age and devote more than 180 min a week to training. Yet several people may meet those conditions but perform different forms of exercise. I found informants such as Carlos, who prepared to run marathons; Gilda, who did not compete but trained six times a week, dividing her workouts between cycling and gym-going; and Alex, who went to the gym almost every day but hardly "trained" and was dedicated to socialising and stretching and eventually lifting some weights. Similarly, criteria such as whether they compete or not, or chronological age, are unsatisfactory because they do not ensure that groups are made up of the same "type" of informant.

Claudia said:

> "I always teach that all bodies are different, all forms of life are different, and each person is a world apart, and that's why I told teachers that before training a person, talk to her, observe her, make her walk, ask her about her medical condition. Age is just a number."

In other words, Claudia advised that trainers who intended to train an older person (or researchers who studied them) should look beyond fixed criteria such as age or hours dedicated to physical exercise, and first and foremost, they should talk to the trainees and understand them in their particularities.

> "Biologically, the doctor told me I'm like a 50-year-old man; imagine, soon I'll be 70. Who do I believe then, the doctor or my ID?" (Francisco)

Francisco often told me that he did not feel like an older adult; as indicated in the quote, this was not his fantasy but was backed up by a statement from a doctor. Cremin's (1992) work is pertinent here; she explained that older adults may distinguish between *being old* and *feeling old*. In other words, age and the marker as older adults are labels that do not represent the reality for some individuals. This qualitative distinction is relevant because, when it is declared that a certain amount of exercise measured in a period is ideal for the older adult population, it must be considered that many people do not feel or identify with the older adult category. Some are in a position to do much more than the amount proposed.

> "There are days and days or even months and months. There are times when I train, and I feel young, energised; there are times when just getting to the gym is an effort. I don't know why, but that's the way it is." (Todd)

Todd introduced more complexity; even the same person can have times when their body allows them to train more or more frequently than usual and times when they can do less than average. The reasons are not motivational, related to discipline or economics, but somewhat random.

### 5.2. The Dangers of Having an Athlete's Identity

> "Everyone who competes knows that competing is at least slightly harmful. My best races were the ones when I had no expectations. When I was focused on winning, those were racing for competition, and I had more injuries, my body ended badly." (Francisco)

For Francisco, sport had to be healthy, and during preparations for competitions or in the competitions themselves, whenever his goal was to win, he mistreated his body. When I asked him to elaborate on this, he said:

> *"For me, it's no longer the goal but the process; my mind is in that mode, and that mentality gives you a level, a standard. If my standard is the podium, I have to assess whether it's worth injuring myself or even doing something that doesn't allow me to run anymore."* (Francisco)

I met Francisco at a stage in his life when he was still competing but no longer intending to win. For him, this was a smart move. He said: "*It's not an age thing, it's not that my body doesn't give me more or I can't be competitive, it's an ego thing. I no longer want to follow the ego; I want to find the balance.*"

Past participation in sports determines current participation (Cousins 1995; Scheerder et al. 2006). Therefore, it is likely that a person who has exercised for a large part of their life will continue to do so as an older adult. These people tend to have an identity (and, as Francisco mentioned, an ego) closely linked to their sporting activity and being an athlete. Nevertheless, studies have found that sports participation in older adulthood can lead to feelings of frustration or fear when performance declines, experiences of negative comments regarding the appropriateness of the person's level of exercise, and contention within their family as a result of the commitment needed to train and compete in sport (Gayman et al. 2017a). These arguments question the implications of ageing identities, athletic injury, and career termination in older athletic populations. They recognise that taking part in sports can lead to unhealthy, maladaptive outcomes such as social withdrawal, depression, difficulty coping, and feelings of guilt, shame or worthlessness (Baker et al. 2010; Winterbotham and du Preez 2016).

With Mara, I talked about workouts, gyms and our expectations of continuing to train even when one day we would be very old. Mara said: "Those who end up suffering are those who don't listen to their body, those who want to be superheroes, those who are still looking to impress others." Mara did not mention the impossibility of training when very old; she did remark that if the training was inadequate, one might end up suffering more than being pleased. For her, the type of training that sought to impress others was of the "superhero" style (and related to what Francisco described as ego).

Although sport is associated with many positive emotions, Grant (2001) noted that it was also a source of frustration when athletes could not perform at previously achieved levels due to changes in functional capacity or physical abilities. Similarly, older people have reported experiencing a fear of ageing due to declining performance results (Eman 2012). Studies have shown that some older athletes put heavy pressure on themselves and feel obligated to maintain their involvement in sports because they fear relinquishing their identity as an athlete (Wigglesworth et al. 2012; Young et al. 2014).

The use of sports to fight ageing may become problematic when individuals deny the inevitable physical decline and disability that is typical of advanced age. Since physical and mental deterioration is unavoidable, these stories of resistance and avoidance may indicate maladaptation to an ageing identity (Baker et al. 2010). Some master athletes have revealed disdain for passive or stereotypical leisure activities and a desire to die on the sports field rather than to live sedentarily (Dionigi 2017).

This is important because many older sports competitors are seen as promoters of successful ageing and as role models for other seniors. Such representations suggest that these groups may be crucial to deconstruct ageist stereotypes (Roters et al. 2010). Nevertheless, studies have found that the achievements of older athletes may reinforce negative stereotypes because they increase prejudice towards physically inactive older adults (Eman 2012; Heo et al. 2013; Pfister 2012). Older adults who exercise tend to differentiate themselves from others by using negative ageing discourse to refer to less active older adults (Sossa 2020). This moralisation of the subject positions active participation in sports as good and those who do not participate in active physical activity may be stigmatised,

victimised, medicalised and ignored in public health policy (Dionigi and O'Flynn 2007; Fullagar 2001; Winterbotham and du Preez 2016).

> *"When cycling, the people who take risks are usually men; I feel the buzz of the bicycle, and what I do is I take space, I stay in that space, I don't cross. I can cause an accident or have one myself, so I don't do that. I enjoy cycling; I go out alone or in groups, but my thing is to enjoy. That's their problem if they want to go faster or show something to others; I feel sorry for them."* (Gilda)

Gilda considered the important thing in cycling to be safety and enjoying the experience. Some research indicates that recreational athletes may experience greater psychological wellbeing than competitive athletes due to higher intrinsic aspirations towards sports participation (Chatzisarantis and Hagger 2007). Studies have reported that older female athletes experience more enjoyment in sports and perceive greater benefits from their participation than their male counterparts (Cardenas et al. 2009). For instance, in a study of men and women of comparable age and types of sports involvement, Eman (2012) reported that women tended to focus on their capabilities and physical experiences and saw themselves as members of an ageing collective, while men tended to find age-related decline in performance to be particularly discouraging and viewed their sporting activities as a way to help slow the process of decline.

*5.3. Are Injuries Inevitable?*

> *"With a lifetime of training, it's obvious that you're going to have some injuries. I have 60% of a tendon torn, and I have tendinosis, plus some osteoarthritis due to age. I feel that it was the gym that caused the injury, but still, when I go to the gym, I feel better than when I don't go."* (Gilda)

This quote summarises several exciting things. First, for Gilda, years of training meant that one was bound to have injuries. Second, despite her injuries caused by gym-going, she continued to go to these venues. Third, even though her training caused her injuries, she reported that her training made her feel good. Lastly, let me compare Gilda's words, regarding men taking more risks and how she felt sorry for them, with this comment: she seemed to naturalise her injury and not attribute it to situations such as training poorly, behaving too riskily or other factors.

Many athletes say that injuries are their best teachers. To some extent, the performance of a beneficial activity also has potential complications. For instance, one study reported that walking effectively prevented hip fractures in men aged 50. However, other studies have suggested that walking increases the risk of fractures, possibly through increased exposure to falls and injury (Holloway-Kew et al. 2018). Is it then better not to walk to avoid exposing yourself to injury? Probably not, but "increasing physical activity will require individuals and groups to walk a fine line between benefit and risk" (Denay et al. 2020, p. 327). It is imperative that adverse effects, intended or unintended, are fully understood when doing physical exercise, as well as the outcomes associated with older adults' involvement in sports (Gayman et al. 2017b).

Many studies have shown that older people who engage in sports activities are convinced they are in better health than those who do not. They think that the more sports they do, the healthier they are, believing that the ageing process can be beaten or postponed via physical effort and determination (Partington et al. 2005). It is central to consider the issue of injuries because, with age, these can be more serious and recovery processes are slower.

> *"One has to know how to do the classes. Older adults don't jump, but some teachers make them jump; there is no reason to do it; it's an unnecessary risk. Injuries are sometimes because of bad luck, it's true, but most of the time, they are due to poorly executed training sessions or poor movements."* (Sandra)

Sandra emphasised that trainers should be well prepared to teach older adults and be aware of the risks for such people as they exercised. However, Callary et al. (2021)

mentioned that coach education programmes included only limited information about the needs of older athletes. This situation is important because the environment that coaches create is critical to the creation of favourable (or deleterious) outcomes in the people they train (Blom et al. 2013; Holt and Neely 2011). High-intensity activities such as jumping may not lead to improved wellbeing (Ryu and Heo 2016).

> *"I've never had injuries because my teachers said my body was born for dancing; it wasn't born to have children. I didn't have children, not because I didn't want to, but because nature acted like that on my body, but I've never had injuries. I have natural elongation, which is educated and ordered with technique and exercise, but it's like when you want to carve something, if you have good wood, you'll be able to carve a sound sculpture; maybe not if the wood is very soft, it won't be as you want, you have to have the element as a base; that's why it's called, the conditions, that's what it's built on. I had the conditions to be a dancer, and I prepared and trained a lot, but always guided, carefully, respecting my body, not hurting it."* (Claudia)

For Claudia, injuries were linked with genetics and not exposing the body to situations that could harm it; like Sandra, she mentioned the importance of performing supervised, guided activities and stated that she had never suffered an injury. Along these lines, when trainers encourage their students to achieve greater levels of physical activity, they should be aware of the increased risk of injuries that result from their trainees being ill-prepared (Bauman et al. 2016; McPhee et al. 2016). There is also the problem of poorly prepared trainers. For this reason, and given that many older adults suffer from functional restrictions (whether or not they are aware of such restrictions) and have limited tolerance for exercise stimuli, exercise warrants safety precautions and careful and appropriate prescriptions (Yamamoto et al. 2022). The role of trained people in guiding this process is vital since incorrect performance can cause severe damage to the body (D'Isanto et al. 2019).

Leinonen et al. (2001) explained that self-rated health was sometimes not a good indicator of objective health, especially among older people. Following a similar logic, we could think that the occurrence of injuries and how to treat them should be a matter not only for the person who suffers them but also for medical specialists. Alex said: "In my medical check-ups, the doctor always asks me if I do sports, to which I answer yes, and he always congratulates me." I asked: "Has the doctor ever asked you what kind of physical activity? And if during the activity, something hurts?" He replied: "Never".

This conversation shows that members of the medical profession assume that sport and physical exercise are good, without any investigation or study of the risks. Only if someone tells a doctor that they have pain, will the doctor ask why. However, it may be beneficial for doctors to evaluate those parts of the body and health that could be compromised based on the physical exercises that a person does, with or without symptoms of pain or injury.

*5.4. The Hazards of Body-Image Ideals*

One last interesting topic is the idea of training and following body-image ideals. Many of my informants trained in gyms, where the young, muscular, toned physique was on show in advertisements; it is a physique that many people aspire to obtain and that coaches tend to possess and encourage clients to have. Added to this trend is that most research on the sport-participation experience and its relationship to the body is focused on young people (Partington et al. 2005). Lean, strong, energetic and youthful concepts are promoted in sports and physical exercise discussions, which tends to leave older adults out of the discourse.

> *"I continuously say in my classes that people look beautiful when they accept themselves. I have 60-year-old students who ask, 'Why should I take care of myself if I'm old?' No, one has always to be beautiful; it's for self-care. Looking good is important, but to look good, you have to feel good first."* (Claudia)

*"I have always struggled with my weight; sports and physical activity were sometimes my enemies because I wanted to lose weight, but when I stopped worrying about my weight, I just started enjoying physical activity."* (Mara)

These quotes tell us that exercise and physical appearance are linked. However, the concern for aesthetics is important only so far as it does not cause problems with how one feels. Similarly, one should not think that because one is a certain age, one no longer has physical beauty. Claudia, as a trainer, and Mara, as a sportsperson, considered that aesthetics should not be the most crucial concern, but accepting one's body was the way to enjoy physical exercise better.

The fitness culture and its members will benefit significantly if they value the physical, recreational, camaraderie, wellbeing and skill aspects at stake in training, especially when the literature shows that individuals are more likely to adopt activities that relevant social groups value when they feel efficient and related concerning those activities (Ryan and Deci 2000).

*"I do classes for older adults focusing on achieving more joint mobility; it's a more functional class than one for younger people. I start with rotations, moving the hands, elbows, everything that has an articulation, and then I go down to the feet, the fingers; some people have arthritis in their hands, some have crooked little fingers. Sometimes, to the rhythm of the music, we do finger massages. Some people have told me that this doesn't make them lose weight, and they complain; I explain that my class isn't for what they want, but for what they need."* (Sandra)

Sandra explained that her classes were not focused on aesthetics but on functional concerns that could help to improve her participants' health. Tokarski (2004) clarified that most programmes for older adults were designed for the "young-old", whereas the "old-old" were left out. Sandra's class seemed to show the opposite; the "too-young" had to adapt to a class that was perhaps less difficult than they expected but more functional for their joints.

*"Once, we were laughing with a friend who had heard that beauty had no age; 'that's what ugly old men say' was our conclusion. Of course, beauty has an age where it flourishes and is seen more. That's why one must understand the times; I train not even thinking about the body, way less about beauty. I train because I feel that my head, memory, and mental agility are better when I exercise."* (Todd).

In other words, Todd did not consider his looks; rather, he trained to help his mind work better. His words are interesting since, often, the words that are used to promote physical exercise focus on body weight, agility and physical rather than mental aspects, whereas, in reality, both are equally important.

## 6. Conclusions

Sport and physical exercise can promote health and wellbeing across the lifespan but are not a panacea (Gayman et al. 2017b). Physical exercise should not be viewed exclusively through a health and performance lens, nor should this lens be considered sacrosanct (Meredith et al. 2023). Virtually all older adults could be physically active, even those with a medical condition, in which case the activity is therapeutic and should be performed to treat the condition (Nelson et al. 2007). For this reason, knowledge and expert guidance are essential to avoid the risks that sports and physical exercise can entail.

In this work, I have shed light on some critical dimensions that ought to be pondered when trainers or researchers consider, plan or study the relationship between physical exercise and older adults. Knowledge of potential benefits and costs can enhance the quality of the ageing experience. Nevertheless, it is essential to emphasise that older adults are not a homogeneous group and that it is highly suggested to define thoroughly what we are talking about when we refer to older adults as well as to physical activity, physical exercise and sports.

The study of sports that are practised later in life is a relatively recent area of investigation, with the majority of work focused on competitive, elite levels of participation (Gayman et al. 2022; Jenkin et al. 2017; Stenner et al. 2020). However, here, I have given voice primarily to non-competitive people who nevertheless train consistently. In this sense, it is necessary to consider and distinguish between those who train for a specific sport and those who train just for the sake of training (as gym-goers usually do). Likewise, it is vital to have research that focuses on those in the older adult population who do not train for competitive purposes. Finally, researchers must know how they train, who guides them, and the risks they may face.

Each person and each body is different, and, at any age, one can be competitive; sports and physical exercise have the characteristics that best help people to feel good. However, my informants considered that involvement in strenuous physical exercise that emphasised winning, triumphing over others and accomplishing challenging physical feats was better for younger people, not older ones. Sports participation can be perceived as enjoyable and a means to build social relationships, but it can also be a source of frustration that results from diminished functional capacity and performance declines (Gayman et al. 2017b); because of this, it is crucial to emphasise the fun, the increased stamina, the discipline and the experience of sports and exercising and not to focus on competition.

As life expectancy increases for many Western populations, the popularity of participating in competitive sports in later life is expected to rise (Baker et al. 2010; Gayman et al. 2017a; Davison and Cowan 2023), and so will the number of non-athletes who like to continue exercising. For this reason, it is vital to understand the qualitative aspects of the relationship between physical exercise and older adults. It is also useful for them to know that sport will not solve every problem and that they should discuss, educate themselves and talk about the possible risks that it entails. Older adults are not a high-priority group for most sporting organisations (Jenkin et al. 2021), but societies are ageing globally, so engaging in this type of discussion is essential.

**Funding:** The study was funded by ANID. Project number: 3220031.

**Institutional Review Board Statement:** The study was conducted in accordance with the Declaration of Helsinki, and approved by the Ethical Committee of the Pontifical Catholic University of Chile. The ethical approval number is 210816008, approved on 11 January 2023.

**Informed Consent Statement:** Informed consent was obtained from all subjects involved in the study.

**Data Availability Statement:** Some data from this research may be shared as required. Other data such as interviews are unavailable due to privacy or ethical restrictions.

**Conflicts of Interest:** The author declares no conflict of interest.

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
