# Peer review of "Physical Exercise and Older People: Always a Happy Relationship? Four Qualitative Reflections to Deepen Understanding"

_socsci, doi:10.3390/socsci13020120_

Round 1
Reviewer 1 Report
Comments and Suggestions for Authors
General Comments
(a) The manuscript does not seem to have the Results and Discussion sections formally labeled/identified. This may be something an editor has already granted permission for, but it does stray from the author guidelines of the journal.
(b) Beyond this, the article was very well-written and does a good job of transitioning smoothly through a wide range of topics. It certainly adds to the available literature regarding sport participation of older adults and addresses some key areas of consideration for practitioners and researchers alike.
Specific Comments
(a) Abstract: Recommend removing use of first person pronouns.
(b) Abstract: The results and conclusion portions of this section (per the journal guidelines) should be added.
(c) Beyond these comments, this manuscript was well-written and was quite easy to read/follow. I see no further necessary changes for the goal of readability and clarity.
Author Response
Dear reviewer, I hope this text finds you well. I want to express my sincere appreciation for your thorough evaluation of my work and valuable feedback.
In relation to your comments and suggestions, I have done the following:
(a) The manuscript does not seem to have the Results and Discussion sections formally labeled/identified. This may be something an editor has already granted permission for, but it does stray from the author guidelines of the journal.
I have added a Results/discussion section.
I also modified the order of some sections, which resulted in a much clearer presentation of my results.
(b) Beyond this, the article was very well-written and does a good job of transitioning smoothly through a wide range of topics. It certainly adds to the available literature regarding sport participation of older adults and addresses some key areas of consideration for practitioners and researchers alike.
Thank you for your comments.
Specific Comments
(a) Abstract: Recommend removing use of first person pronouns.
I have kept using first-person pronouns since it is more consistent with the study's logic and the methodological tradition used, namely, ethnographic work.
(b) Abstract: The results and conclusion portions of this section (per the journal guidelines) should be added.
I have based my abstract on the latest articles published by the journal; those texts do not have these sections. I have chosen to leave the writing of the abstract as it is since it has a fluidity more in line with the qualitative works that I discuss in my work.
(c) Beyond these comments, this manuscript was well-written and was quite easy to read/follow. I see no further necessary changes for the goal of readability and clarity.
Thanks again for your feedback.
I want to let you know that all the revised sections and additional changes have been clearly marked in the manuscript for the convenience of the reviewers and editors.
Reviewer 2 Report
Comments and Suggestions for Authors
This study gives voice to aging populations and their orientations to sport, physical activity, and exercise. The qualitative nature of this study is useful for providing rich information regarding these perspectives and offers some direction for multiple fields regarding conceptualizing recommendations and discussion of these topics with this group. There are several ways to reorganize this manuscript to help better communicate these findings listed below.
Introduction
· Lines 25-29 are a bit confusing. It sounds as though the authors are suggesting sport, physical exercise, and physical activity are not defined, which is outside the scope of this manuscript. I would suggest a rewrite here of something similar to: “However, the literature does not clarify two critical elements as it relates to older populations. Firstly, what do we mean by sport, physical exercise and physical activity, in relation to these populations? Second, what margins and precautions should older people take, and what damage can physical exercise cause to this population?”
· Line 53 notes “beat”-ing others or as a benefit of sport, but there is significant research suggesting that focusing on competition may actually hinder benefits of physical activity on mental health. With that in mind, I believe it would make more sense to say something similar to, “enhanced connectedness to the present moment through task-focused engagement” since there is substantial literature detailing the positive benefits of present moment awareness. Another way to frame this could be through the associations with perceived competence or self-efficacy.
· It seems a bit like a case is being made at the end of the introduction about drawing distinctions between the benefits of sport, physical activity, and exercise, but that is not presented from the outset. I think this could be included as one of the critical elements early in the introduction to help set the stage for the study.
Methodology
· The first few paragraphs of the methodology section seem to be more focused on introducing the study as opposed to detailing the methodology of the study. I believe it would be best to have this moved to the Introduction. These paragraphs should also be adjusted to focus more on the empirical support as opposed to the lived experience of the researcher. This latter part can be used, but there needs to be a better integration of how this experience ties into the literature throughout this portion.
· Broadly speaking, the lack of details regarding which studies the participants were pulled from and what type of setting the conversations introduces potential areas of bias that would need more clarity to ensure the bias is mitigated. Although there might be some of that information contained in the blinded citations, please bring some orientation to that in the text of the manuscript.
· Please clarify somewhere in this section that the names of those providing quotes for the studies are not the real names of the participants.
Results and Discussion
· The first two sections (i.e., “Some Definitions Problematized” and “Some Adverse Outcomes of Exercise and Sport”) seem like they fit better in the introduction section as opposed to the results as they provide information about the nature of the endeavor as opposed to detailing information gleaned from the study.
· The journal submission guidelines indicate the following section headers should be included: “Introduction, Materials and Methods, Results, Discussion, Conclusions (optional).” The current headers could be included as subheadings, and a Results/Discussion section could be added since the findings and interpretations of findings flow synergistically given the qualitative nature of the study. However, there also seems to be a distinction between the qualitative data collected and the interpretations of those findings by the author, so I would suggest that these be made separate in a Results and Discussion section, respectively.
· In line 436, Claudia is italicized, but the names of the participants are not italicized in other quotes. Please change for consistency.
· There are several themes present that are consistent with Deci and Ryan’s Self Determination Theory or the Social Motivational Orientations in Sport Scale (SMOSS; Allen, 2005 tested the initial construct validity) and more recent studies have examined the factor structure. It could be helpful to review some of this literature and integrate it into the Discussion, particularly the deleterious effects of appearance-related motivations for exercise.
Conclusion
· The author frequently uses “should” throughout the conclusion, and it would seem more aligned with the nature of the study to instead use phrases such as “it is highly suggested” or “it could be impactful.”
· The author notes that life expectancy is increasing for many western cultures in line 507, but there has actually been a decline in many western cultures in recent years (2020 and 2021). Although it may have gone up since then, I have not seen that data, and a citation is warranted for this claim.
Author Response
Dear reviewer, I hope this text finds you well. I want to express my sincere appreciation for your thorough evaluation of my work and valuable feedback. Your comments have immensely contributed to improving the quality and clarity of my article.
In relation to your comments and suggestions, I have done the following:
This study gives voice to aging populations and their orientations to sport, physical activity, and exercise. The qualitative nature of this study is useful for providing rich information regarding these perspectives and offers some direction for multiple fields regarding conceptualizing recommendations and discussion of these topics with this group. There are several ways to reorganize this manuscript to help better communicate these findings listed below.
Introduction
- Lines 25-29 are a bit confusing. It sounds as though the authors are suggesting sport, physical exercise, and physical activity are not defined, which is outside the scope of this manuscript. I would suggest a rewrite here of something similar to: “However, the literature does not clarify two critical elements as it relates to older populations. Firstly, what do we mean by sport, physical exercise and physical activity, in relation to these populations? Second, what margins and precautions should older people take, and what damage can physical exercise cause to this population?”
I have changed the sentence as suggested.
- Line 53 notes “beat”-ing others or as a benefit of sport, but there is significant research suggesting that focusing on competition may actually hinder benefits of physical activity on mental health. With that in mind, I believe it would make more sense to say something similar to, “enhanced connectedness to the present moment through task-focused engagement” since there is substantial literature detailing the positive benefits of present moment awareness. Another way to frame this could be through the associations with perceived competence or self-efficacy.
In the text, you will find that I have changed the sentence following your feedback.
- It seems a bit like a case is being made at the end of the introduction about drawing distinctions between the benefits of sport, physical activity, and exercise, but that is not presented from the outset. I think this could be included as one of the critical elements early in the introduction to help set the stage for the study.
I have added some sentences and changed the article's structure, advancing some sections to better prepare the results and discussion section.
Methodology
- The first few paragraphs of the methodology section seem to be more focused on introducing the study as opposed to detailing the methodology of the study. I believe it would be best to have this moved to the Introduction. These paragraphs should also be adjusted to focus more on the empirical support as opposed to the lived experience of the researcher. This latter part can be used, but there needs to be a better integration of how this experience ties into the literature throughout this portion.
This is an excellent suggestion that made me rethink whether or not to include these lines in the introduction. In the end, I have chosen to leave it in the methodology because although these paragraphs do not introduce the study's methodology, they do introduce the researcher, who, in ethnographic studies, is very significant since it is “the tool” for collecting data.
Now, since this does not solve the problem, I have shortened these paragraphs and rewritten them better so that they are in better harmony with the methodological section of the study.
- Broadly speaking, the lack of details regarding which studies the participants were pulled from and what type of setting the conversations introduces potential areas of bias that would need more clarity to ensure the bias is mitigated. Although there might be some of that information contained in the blinded citations, please bring some orientation to that in the text of the manuscript.
It's a good observation. Therefore, I added one more work to the bibliography and an explanatory sentence in the methodology section. With this, if someone wants to corroborate or know more about the people mentioned, they can do so more efficiently.
- Please clarify somewhere in this section that the names of those providing quotes for the studies are not the real names of the participants.
In the previous version, it was already mentioned that the data was anonymised. Nevertheless, With the sentence added to the methodology, this becomes even clearer.
Results and Discussion
- The first two sections (i.e., “Some Definitions Problematized” and “Some Adverse Outcomes of Exercise and Sport”) seem like they fit better in the introduction section as opposed to the results as they provide information about the nature of the endeavor as opposed to detailing information gleaned from the study.
I appreciate your comments regarding the section headers. These can sometimes confuse more than guide. I have moved the sections pointed out as sections after the introduction. I have not included them as part of the introduction, as this would make it too long and confusing.
- The journal submission guidelines indicate the following section headers should be included: “Introduction, Materials and Methods, Results, Discussion, Conclusions (optional).” The current headers could be included as subheadings, and a Results/Discussion section could be added since the findings and interpretations of findings flow synergistically given the qualitative nature of the study. However, there also seems to be a distinction between the qualitative data collected and the interpretations of those findings by the author, so I would suggest that these be made separate in a Results and Discussion section, respectively.
As you mentioned, the findings and interpretations of findings flow synergistically, given the qualitative nature of the study. For this reason, I have added the discussion/results header (but together so as not to dismember how this section has been written).
- In line 436, Claudia is italicized, but the names of the participants are not italicized in other quotes. Please change for consistency.
Thank you for noticing this. I have made the corresponding changes to maintain consistency.
- There are several themes present that are consistent with Deci and Ryan’s Self Determination Theory or the Social Motivational Orientations in Sport Scale (SMOSS; Allen, 2005 tested the initial construct validity) and more recent studies have examined the factor structure. It could be helpful to review some of this literature and integrate it into the Discussion, particularly the deleterious effects of appearance-related motivations for exercise.
It is a good recommendation, and I have included one text:
Ryan, R. M., & Deci, E. L. (2000). Self-determination theory and the facilitation of intrinsic motivation, social development, and well-being. American Psychologist, 55(1), 68–78. https://doi.org/10.1037/0003-066X.55.1.68
I have not included more because this literature tends to dialogue better with psychological or psychosocial theory and less with anthropology or sociology, which are the disciplines that I have cited the most in this study.
Conclusion
- The author frequently uses “should” throughout the conclusion, and it would seem more aligned with the nature of the study to instead use phrases such as “it is highly suggested” or “it could be impactful.”
It is a very good suggestion and I have made the modifications.
- The author notes that life expectancy is increasing for many western cultures in line 507, but there has actually been a decline in many western cultures in recent years (2020 and 2021). Although it may have gone up since then, I have not seen that data, and a citation is warranted for this claim.
I have corrected this by adding a study that presents this data.
Thanks again for your feedback.
I want to let you know that all the revised sections and additional changes have been clearly marked in the manuscript for the convenience of the reviewers and editors.
Round 2
Reviewer 2 Report
Comments and Suggestions for Authors
Thank you for your attentiveness to these changes. The manuscript now seems to be sufficiently improved for publication.